# Memory Architectures in Recurrent Neural Network Language Models

**Dani Yogatama♣, Yishu Miao♠, Gabor Melis♣, Wang Ling♣, Adhiguna Kuncoro♣♠**
**Chris Dyer♣, Phil Blunsom♣♠**
♣DeepMind and ♠University of Oxford
dyogatama@google.com, yishu.miao@cs.ox.ac.uk
{melisgl,lingwang,akuncoro,cdyer,pblunsom}@google.com

## Abstract

We compare and analyze sequential, random access, and stack memory architectures for recurrent neural network language models. Our experiments on the Penn Treebank and Wikitext-2 datasets show that stack-based memory architectures consistently achieve the best performance in terms of held out perplexity. We also propose a generalization to existing continuous stack models (Joulin & Mikolov, 2015; Grefenstette et al., 2015) to allow a variable number of pop operations more naturally that further improves performance. We further evaluate these language models in terms of their ability to capture non-local syntactic dependencies on a subject-verb agreement dataset (Linzen et al., 2016) and establish new state of the art results using memory augmented language models. Our results demonstrate the value of stack-structured memory for explaining the distribution of words in natural language, in line with linguistic theories claiming a context-free backbone for natural language.

## 1 Introduction

Sequential recurrent neural networks such as LSTMs (Hochreiter & Schmidhuber, 1997) are the basis of state-of-the-art models of natural language in various tasks. They effectively learn to capture dependencies between events separated in time by learning to store and retrieve information in a hidden state. However, the ability of these methods to discover long-term dependencies is limited by the capacity of their hidden state and the difficulty of propagating reliable gradients. For example, LSTM language models have been shown to struggle to capture non-sequential syntactic dependencies in complex sentences without explicit supervision (Linzen et al., 2016). As an illustration of the kind of dependencies they have difficulty learning, in the sentence, *the **loss** of basic needs providers emigrating from impoverished countries **has** a damaging effect*, correctly predicting singular *has* rather than its plural form *have* requires that the LSTM have learned that it depends on the subject, in this case the first noun (*loss*) rather than any of the intervening non-subject nouns, such as *countries*. Linzen et al. (2016) show that LSTM language models fail to capture this kind of dependencies, especially as the number of attractors (underlined) increases.

Attempts to improve language models' ability to capture non-local dependencies have recently been undertaken by introducing an external memory components. These include (i) a soft attention mechanism (Daniluk et al., 2017) and (ii) an explicit memory block or cache model (Tran et al., 2016; Grave et al., 2017). However, since very local context is often most highly informative for predicting the next word, existing memory-augmented RNN LMs use memory just to store information about local context (Daniluk et al., 2017).

In this work, we compare several memory architectures for recurrent neural network language models. Since our goal is to evaluate how well these types of memory architectures learn long term and syntactic dependencies, we focus on language models that are static as opposed to non-static models such as neural cache (Grave et al., 2017) and dynamic evaluation (Krause et al., 2017) that can update their distribution at *test time*. We consider increasing the capacity of a purely sequential memory model by increasing the capacity of an LSTM, a random access memory model as typified by an attention-based LSTM, and a new variant of a stack augmented recurrent neural network.

Unlike random access memory models, a stack has a built-in bias to discover hierarchical structures that are important in language. A continuous stack memory has been proposed to improve recurrent neural networks (Joulin & Mikolov, 2015; Grefenstette et al., 2015), although it has never been carefully evaluated in benchmark language modeling experiments. In the only set of results for a stack augmented recurrent language model, Joulin & Mikolov (2015) show that a stack augmented vanilla RNN outperforms a standard RNN and is comparable to an LSTM. We augment an LSTM with a stack memory and perform thorough comparisons to evaluate its efficacy as a language model. In contrast to prior work, our continuous stack allows for push, stay, and a variable number of pop operations at each time step (multiple pop operations are useful in modeling natural language sentences since while only a single new word is presented at each time step, multiple syntactic units may come to an end concurrently).

Our motivating hypothesis is that allowing the memory to dynamically store and retrieve contextual information with a stack will drive the model to use the memory to learn dependencies that are difficult to capture by a sequential model. Sequential memory has an easier time learning local dependencies, but it often fails to capture long term dependencies. Random access memory models capture longer range dependencies (i.e., proportional to the window size), but the learner has to infer these from data without any informative structural bias. We hypothesize that introducing a more appropriate inductive bias will make it easier for the model to learn long range and structurally meaningful dependencies, given that the variance in learning such dependencies can be high. Linguistic insights reveal that one possible inductive bias is in the form of a hierarchical nested structure that captures syntactic dependencies. Stack memory models provide a natural way for capturing hierarchical structures, providing an easier path for gradients to flow to particular locations in the past.

Our main contributions in this paper are as follows:

- We thoroughly evaluate the efficacy of a stack augmented RNN as a language model and propose a more expressive extension to existing stack models (§2.1).
- We compare how a recurrent neural network uses a stack memory, a sequential memory cell (i.e., an LSTM memory cell), and a random access memory (i.e., an attention mechanism) for language modeling. Experiments on the Penn Treebank and Wikitext-2 datasets (§3.2) show that both the stack model and the attention-based model outperform the LSTM model with a comparable (or even larger) number of parameters, and that the stack model eliminates the need to tune window size to achieve the best perplexity.
- We assess the ability of these memory models to discover long range structural dependencies commonly encountered in natural language using the subject-verb agreement dataset (Linzen et al., 2016). We achieve new state of the art results and show that the gap in accuracy between a sequential or random access memory model with a stack model gets bigger as the dependencies become more complex (i.e., number of attractors increases; §3.3). We also analyze the stack and find that the model tends to use it to enhance its sequential memory component in high entropy prediction contexts (§3.4).

## 2 MODEL

We consider a language modeling problem where the goal is to predict the next word $x_t$ given previously seen context words $x_0, \ldots, x_{t-1}$. We represent each input word $x$ by its $D$-dimensional embedding vector $\mathbf{x} \in \mathbb{R}^D$.

Our base model is an LSTM that computes a hidden state at timestep $t$ as follows:

$$\mathbf{i}_t = \sigma(\mathbf{W}_{i,x}\mathbf{x}_t + \mathbf{W}_{i,h}\mathbf{h}_{t-1} + \mathbf{b}_i) \qquad \mathbf{f}_t = \sigma(\mathbf{W}_{f,x}\mathbf{x}_t + \mathbf{W}_{f,h}\mathbf{h}_{t-1} + \mathbf{b}_f)$$
$$\mathbf{o}_t = \sigma(\mathbf{W}_{o,x}\mathbf{x}_t + \mathbf{W}_{o,h}\mathbf{h}_{t-1} + \mathbf{b}_o) \qquad \mathbf{g}_t = \tanh(\mathbf{W}_{g,x}\mathbf{x}_t + \mathbf{W}_{g,h}\mathbf{h}_{t-1} + \mathbf{b}_g)$$
$$\mathbf{c}_t = \mathbf{f}_t \odot \mathbf{c}_{t-1} + \mathbf{i}_t \odot \mathbf{g}_t \qquad \mathbf{h}_t = \mathbf{o}_t \odot \tanh(\mathbf{c}_t)$$

**Sequential memory.** An LSTM has a sequential memory cell $\mathbf{c}$ to store and retrieve information that is regulated by the input, output, and forget gates. In order for long-term contextual information to be used in the future, it has to pass through these gates for multiple timesteps.

**Random access memory.** One common approach to retrieve information from the distant past more reliably is to augment the model with a random access memory block via an attention based

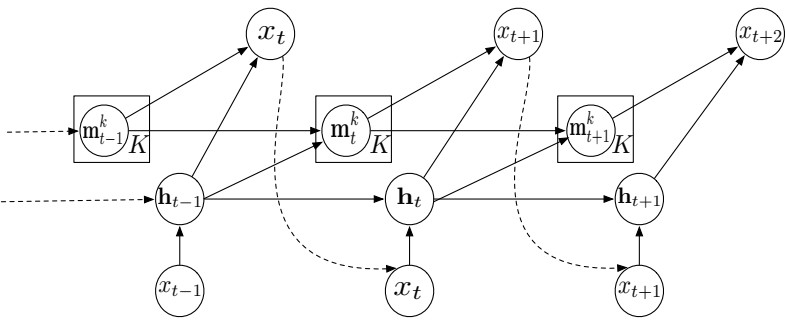

Figure 1: Multipop Adaptive Computation Stack Recurrent Neural Network.

method. In this model, we consider the previous $K$ states as the memory block, and construct a memory vector $\mathbf{m}_t$ by a weighted combination of these states:

$$\mathbf{m}_t = \sum_{i=t-K}^{t-1} a_i \mathbf{h}_i, \text{ where } a_i \propto \exp(\mathbf{w}_{m,i}\mathbf{h}_i + \mathbf{w}_{m,h}\mathbf{h}_t)$$

Such method can be improved further by partitioning $\mathbf{h}$ into a key, value, and predict subvectors (Daniluk et al., 2017).

Given the LSTM hidden state $\mathbf{h}_t$ and the memory state $\mathbf{m}_t$, we combine them using a simple function:

$$\tilde{\mathbf{h}}_t = \mathbf{W}_{h,h}\mathbf{h}_t + \mathbf{W}_{h,m}\mathbf{m}_t$$

to get the final representation $\tilde{\mathbf{h}}_t$. We compute the probability of predicting the next word as $p(x_t \mid \boldsymbol{x}_{<t}) \propto \exp(\mathbf{x}_t^\top \tilde{\mathbf{h}}_t + b_{y,x_t})$, where we follow Inan et al. (2017) and reuse the word embedding matrix $\mathbf{X}$ as the softmax parameters.

**Stack memory.** In this work, we propose to augment a recurrent LSTM language model with a stack memory $\mathbf{M}$ that has three basic operations:

- PUSH: Push the current hidden state $\mathbf{h}_t$ onto the stack.
- POP: Remove the top element of the stack.
- STAY: Keep the stack unchanged.

Figure 1 shows an illustration of our stack augmented RNN. We describe the stack memory in details in the followings.

## 2.1 MULTIPOP ADAPTIVE COMPUTATION STACK

Our stack is a multipop adaptive computation stack—it learns how many POP operations need to be performed before predicting an output. In previous work (Joulin & Mikolov, 2015; Grefenstette et al., 2015), at every timestep $t$, the job of the memory (stack) controller is to decide whether (i) to push the current state (either $\mathbf{h}_t$ or $\mathbf{x}_t$) onto the stack, (ii) to pop the top element of the stack $\mathbf{m}_0$, or (iii) to stay and keep the stack state unchanged. The stack of Joulin & Mikolov (2015) is primarily designed as a single computation stack that performs one of the available operations at every timestep. In order to capture long-term dependencies, the stack learns to carry the information across multiple timesteps by mainly relying on the LSTM hidden states for predictions in between and keeping the state of the stack the same (i.e., by choosing to stay), or by pushing and popping the same number of times in between these timesteps. While this promotes discoveries of hierarchical dependencies, the kind of hierarchical dependencies that it can discover is limited. A multipop stack, on the other hand, has greater flexibility since there are more ways to manipulate its state at each timestep. The stack of Grefenstette et al. (2015) implicitly allows multiple pop operations in a single timestep by setting the pop weights to be greater than one. However, the controller makes this decision based only on the element at the top of the stack (along with the input and the current hidden state), making

it less plausible to know whether more than one pop operations are needed since it does not look at other elements of the stack. Our formulation of the multipop operations is more intuitive and takes inspirations from adaptive computation time (Graves, 2017).

Concretely, consider a stack memory with $K$ elements. In all our experiments, we limit the size of the stack to $K = 10$ for computational considerations. If the stack requires more than $K$ elements, the bottom element of the stack is removed to make space for the new element, which is added on the top of the stack. In a single computation stack, a feedforward policy network is used to compute the probability of choosing an action $a \in \{\text{STAY}, \text{PUSH}, \text{POP}\}$.

In our stack, we also use a feedforward policy network, but the number of possible POPs is $k \in \{0, 1, \ldots, K\}$. Denote the current top two elements of the stack after performing $k$ pops by $\mathbf{m}_{k,0}$ $\mathbf{m}_{k,1}$, and the state of the stack after $k$ pops by $\text{STAY}_k$ (i.e., do $k$ pops and stay) or $\text{PUSH}_k$ (i.e., do $k$ pops and push the current hidden state $\mathbf{h}_t$). We compute the probability of choosing an action recursively:

$$p(\text{STAY}_k \mid \mathbf{x}_t, \mathbf{M}) = p(\text{POP}_{k-1} \mid \mathbf{x}_t, \mathbf{m}_{k-1,0}, \mathbf{m}_{k-1,1}) \times p(\text{STAY}_k \mid \mathbf{x}_t, \mathbf{m}_{k,0}, \mathbf{m}_{k,1})$$
$$p(\text{PUSH}_k \mid \mathbf{x}_t, \mathbf{M}) = p(\text{POP}_{k-1} \mid \mathbf{x}_t, \mathbf{m}_{k-1,0}, \mathbf{m}_{k-1,1}) \times p(\text{PUSH}_k \mid \mathbf{x}_t, \mathbf{m}_{k,0}, \mathbf{m}_{k,1})$$
$$p(\text{POP}_k \mid \mathbf{x}_t, \mathbf{M}) = p(\text{POP}_{k-1} \mid \mathbf{x}_t, \mathbf{m}_{k-1,0}, \mathbf{m}_{k-1,1}) \times p(\text{POP}_k \mid \mathbf{x}_t, \mathbf{m}_{k,0}, \mathbf{m}_{k,1}).$$

As a base case, we have $p(\text{POP}_{-1} \mid \mathbf{x}_t, \mathbf{m}_{k-1,0}, \mathbf{m}_{k-1,1}) = 1$. To ensure that the probability sums to one, we set $p(\text{POP}_{K+1} \mid \mathbf{x}_t, \mathbf{m}_{K,0}, \mathbf{m}_{K,1}) = 0$.

The final stack state is then computed as:

$$\mathbf{M} = \sum_{k=0}^{K} p(\text{STAY}_k \mid \mathbf{x}_t, \mathbf{M})\mathbf{M}_{\text{STAY}_k} + p(\text{PUSH}_k \mid \mathbf{x}_t, \mathbf{M})\mathbf{M}_{\text{PUSH}_k}, \tag{1}$$

where $\mathbf{M}_{*_k}$ is the stack state after performing $k$ POP and PUSH or STAY. Denote the top of the final stack at timestep $t$ as $\mathbf{m}_t$. The final representation is

$$\tilde{\mathbf{h}}_t = \mathbf{W}_{h,h}\mathbf{h}_t + \mathbf{W}_{h,m}\mathbf{m}_t.$$

We propose to treat this stack as a fully differentiable continuous stack. Alternatively, our stack can also be treated as a discrete stack and trained with reinforcement learning (e.g., with REINFORCE; Williams, 1992). In this case, instead of summing over all possible stack states, we sample according to the probabilities. However, such methods can have slow convergence due to high variance. We include comparisons to discrete and continuous single computation stacks in our experiments (§3).

**Adaptive and Variable Computation Networks**   Previous work on adaptive computation time (Graves, 2017) consider the number of computations as "thinking time", where they show that their models use more computation time for more difficult predictions. In our work, the number of computations is related to how further back we need to look back when making a prediction at a given timestep. Note that when we decide to push, we push the current hidden state $\mathbf{h}_t$ onto the stack. Since this operation is performed before making a prediction at every timestep, it is possible to use the stack to increase the number of parameters for some predictions (i.e., by pushing $\mathbf{h}_t$ and immediately use it to compute $\tilde{\mathbf{h}}_t = \mathbf{W}_{h,h}\mathbf{h}_t + \mathbf{W}_{h,m}\mathbf{m}_t$, because immediately after a push $\mathbf{m}_t = \mathbf{h}_t$). As a result, our stack is also related to variable computation recurrent networks Jernite et al. (2017) that decide the number of dimensions to be used at each timestep.

## 3 EXPERIMENTS

### 3.1 SETUP

We compare the following methods in our experiments:

- Sequential memory: 650-dimension and 920-dimension vanilla LSTMs (650 or 920 for both the word embedding and the LSTM hidden size).
- Random access memory: a 650-dimension attention-based LSTM with attention size $K = \{1, 3, 5, 10, 15\}$.

- Stack memory: a single computation discrete or continuous stack, or a multipop adaptive computation continuous stack on top of a 650-dimension LSTM.

Following Inan et al. (2017), we tie word embedding and word classifier layers and apply dropout to these layers with probability 0.6 (value chosen based on preliminary experiment results). We also use recurrent dropout (Semeniuta et al., 2016) and set it to 0.1. We perform non-episodic training with batch size 32 using RMSprop (Hinton, 2012) as our optimization method. We tune the RMSprop learning rate and $\ell_2$ regularization parameter for all models on a development set by random search from $[0.004, 0.009]$ and $[0.0001, 0.0005]$ respectively, and use perplexity on the development set to choose the best model.

## 3.2 PERPLEXITY

We use standard language modeling datasets, the Penn TreeBank (PTB) and Wikitext-2 (Wik-2) corpora to evaluate perplexity. Our main results are summarized in Table 1, where we also show comparisons with previous work on these datasets.

Our basic LSTM model is comparable to some of the best LSTM models. The results show that increasing the sequential memory capacity by increasing the hidden size improves performance. However, the improvement is not as significant as adding random access or stack memory. The best attention model is the one with $K = 10$ and $K = 15$ on PTB and Wik-2 respectively, highlighting the necessity to tune to get the optimal window size. Our results generally agree with Daniluk et al. (2017) that show that increasing the attention size generally improves performance up to a certain threshold.

While both the discrete and continuous single computation stack models perform reasonably well, the discrete model underperforms the continuous model on Wik-2. Recall that we use REINFORCE to learn the optimal discrete stack operations. We leave it to future work to investigate whether better techniques can be used to improve the performance of the discrete stack. The best model on both datasets is consistently the multipop stack.

Overall perplexity on these datasets is strongly dominated by words that require little to no long term dependencies, making it difficult to assess when memory helps. In the next section, we look into a specifically designed linguistic task to get a better understanding of these memory models.

## 3.3 SYNTACTIC DEPENDENCIES

We evaluate these memory models for learning syntax-sensitive dependencies on the number prediction dataset from Linzen et al. (2016). In this dataset, the model is given a sentence up to—but not including—its verb, and the goal is to predict the number of the following verb (singular or plural). For example, given a sentence prefix with different numbers of intervening nouns:

- *this **robot** {**is**, are}*

- *the **users** he mentioned {is, **are**}*

- *many **systems** , in addition to VBG a page of free text for each knowledge element , also {**permit**, permits}*

the goal is to predict the correct verb form out of the possible answers in the brackets. In total, there are approximately 1.4 million test examples in this dataset with varying degrees of difficulty. One proxy to assess the difficulty of a test example is through the number of **attractors** (underlined)—which are defined as intervening nouns of the opposite singular/plural form to the subject. Each of the example above has zero, one, and four attractors, respectively. Naturally, examples with fewer numbers of attractors between the head of the syntactic subject and the predicted verb are easier than those with more. We follow the experimental setup in Linzen et al. (2016) and only use test examples where *all* the attractors are of contrasting form to the main subject (i.e., all intervening nouns between the subject and the verb must be plural if the subject is singular, and vice versa).

One way to do this task is to train a binary classifier that takes the context and predicts an answer. We approach this task from a language modeling perspective, where we simply train a language model

| Model | LSTM hidden size | # of params. | PTB | | Wik-2 | |
|---|---|---|---|---|---|---|
| | | | Dev | Test | Dev | Test |
| Var LSTM (Gal & Ghahramani, 2016) | - | 20M | 81.9 | 79.7 | 101.7 | 96.3 |
| Var LSTM+REAL (Inan et al., 2017) | 1500 | 51M | 71.1 | 68.5 | - | - |
| Pointer LSTM (Merity et al., 2017b) | - | 21M | 72.4 | 70.9 | 84.8 | 80.8 |
| Neural Cache (Grave et al., 2017) | - | - | - | 72.1 | - | 81.6 |
| Neural Cache (Grave et al., 2017) | - | - | - | - | - | 68.9 |
| NAS (Zoph & Le, 2017) | - | 54M | - | 62.4 | - | - |
| Optimized LSTM (Melis et al., 2017) | - | 24M | 60.9 | 58.3 | 69.1 | 65.9 |
| AWD LSTM (Merity et al., 2017a) | - | 24M | 60.0 | 57.3 | 68.6 | 65.8 |
| AWD LSTM + Cache (Merity et al., 2017a) | - | 24M | 53.9 | 52.8 | 53.8 | 52.0 |
| LSTM | 650 | 10M/25M | 69.2 | 67.2 | 83.9 | 80.8 |
| LSTM | 920 | 16M/40M | 67.8 | 65.4 | 79.8 | 77.4 |
| Attention-1 | 650 | 12M/28M | 68.6 | 66.1 | 80.4 | 76.3 |
| Attention-3 | | | 67.9 | 65.4 | 78.7 | 74.6 |
| Attention-5 | | | 67.5 | 65.2 | 78.2 | 74.6 |
| Attention-10 | | | 67.2 | 64.7 | 77.6 | 73.7 |
| Attention-15 | | | 66.6 | 63.6 | 77.7 | 74.3 |
| Single Comp. Discrete Stack | 650 | 11M/26M | 66.1 | 63.5 | 78.1 | 74.7 |
| Single Comp. Continuous Stack | | | **65.8** | 63.8 | 76.7 | 73.0 |
| Multipop Adaptive Continuous Stack | | | 65.9 | **63.5** | **75.9** | **72.4** |

Table 1: Perplexity on PTB and Wikitext-2 datasets. The two numbers (*M/*M) in the # of params. column for models that we implemented denote the number of parameters for PTB and Wik-2 respectively.

and take the word with the higher probability between the two possible answers as the prediction. Success on this task requires a language model that understands syntactic—and in some cases long term—dependencies in natural language. For a purely sequential memory model to do well on this task, it has to be able to carry dependencies over multiple timesteps and attractors. On the other hand, our memory augmented recurrent models need to use the random access or stack memory component in conjunction with the sequential memory of their LSTM core to capture these dependencies.

Linzen et al. (2016) concluded that a vanilla LSTM trained only with a language modeling signal is insufficient for capturing such dependencies. They reported an overall accuracy of 93.22 with a language modeling objective (using a 50 dimension LSTM), and 99.17 with a supervised binary classifier objective.

We train the best vanilla LSTM (920 dimensions), the best attention-based LSTM ($K = 10$, since the Linzen dataset is derived from Wikipedia articles similar to Wik-2), and the best stack LSTM (multipop stack) on the provided training set that contains sentences of similar structures to the test set. There are approximately 3 million tokens on the training set ($\sim$140,000 sentences). We tune the learning rate and $\ell_2$ hyperparameter on the development set using perplexity as the tuning criterion.

Our results are shown in Table 2. We report accuracies per number of attractors, as well as the overall accuracy and perplexity. Contrary to the Linzen et al. (2016) results, all of our language models perform surprisingly well on this dataset. Our vanilla LSTM model outperforms Linzen's best LSTM by a significant margin (99.11 vs. 93.22). One possible reason is that we are able to train a much bigger LSTM than Linzen—almost 20 times bigger in hidden size. The results clearly demonstrate the improvements from adding random access and stack memory. The performance of the attention model slowly degrades to the performance of a vanilla LSTM model as the number of attractors increases, since it becomes more difficult for a random access memory mechanism to attend to the syntactic head in the presence of multiple attractors. For example, when there are five attractors, our attention model performs just as well as our vanilla LSTM model.

| Model | Number of attractors | | | | | | Acc. | Ppx. |
|---|---|---|---|---|---|---|---|---|
| | **0** | **1** | **2** | **3** | **4** | **5** | | |
| Best LSTM | 99.3 | 97.2 | 95.0 | 92.2 | 90.0 | 84.2 | 99.11 | 23.8 |
| Best attention | **99.4** | 97.7 | 95.9 | 92.9 | 90.7 | 84.2 | 99.18 | 22.7 |
| Best stack | **99.4** | **97.9** | **96.5** | **93.5** | **91.6** | **88.0** | **99.23** | **22.2** |

Table 2: Accuracies on the Linzen number prediction dataset. 0, 1, 2, 3, 4, and 5 refer to the number of attractors between the subject and the predicted verb (see text for details).

| Model | | | Example |
|---|---|---|---|
| LSTM | attention | stack | |
| ✗ | ✗ | ✗ | *the NN notes and front cover title {is,**are**}* |
| ✗ | ✗ | ✓ | *other NNS that in the recent past were part of the JJ parish {is,**are**}* |
| ✗ | ✓ | ✗ | *the class of all VBN sets with JJ functions as NNS {form,**forms**}* |
| ✓ | ✗ | ✗ | *various brands of JJ compound or NN NN {helps,**help**}* |
| ✗ | ✓ | ✓ | *score based on penalties for fallen bars , NNS , {**falls**,fall}* |
| ✓ | ✗ | ✓ | *the loss of basic needs providers VBG from VBN countries {**has**,have}* |
| ✓ | ✓ | ✗ | *the construction of the JJ walls , floors , and VBG walls {**is**,are}* |

Table 3: Examples of mistakes made by competing models on the Linzen number prediction dataset. ✗ indicates an incorrect prediction, whereas ✓ indicates a correct prediction. In general, we observe that the mistakes made by both the LSTM and attention models that are correctly predicted by the stack model (row 2) typically involve longer sentences regardless of the number of attractors.

The stack model performs best on this dataset, across all numbers of attractors (except zero, tie with attention), Notably, the advantage of the stack model becomes more pronounced as the number of attractors increases. In §3.4, we analyze how the model uses its stack. We take this collection of results as evidence that a hierarchical bias introduced by a stack-like data structure helps the language model to learn better syntactic natural language dependencies.

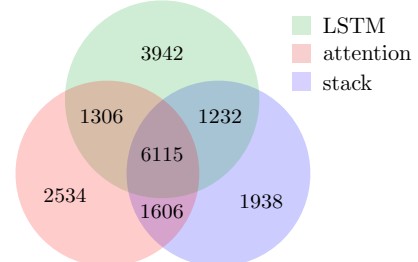

Figure 2: A Venn diagram of mistakes made on the Linzen dataset.

We also investigate whether mistakes are made on the same test examples. Figure 2 shows a Venn diagram of mistakes made by each of the models. Most mistakes (6115) are the same across all three models. It is clear that adding a stack or an attention mechanism improves a vanilla LSTM model, as shown by the significant decrease in the number of mistakes that are made only by LSTM (3942) to 1938 and 2534 respectively. Nonetheless, since there are still a large number of mistakes that are complementary, an interesting future direction is to combine all three kinds of memory models efficiently in a single language model. Table 3 shows examples of mistakes made by each of these models.

## 3.4 ANALYSIS

In this section, we analyze how our stack model uses its memory to improve predictions. Recall that our adaptive continuous stack has 22 possible stack states at each timestep—$\text{STAY}_0, \text{PUSH}_0, \text{STAY}_1, \text{PUSH}_1, \ldots, \text{STAY}_{10}, \text{PUSH}_{10}$—where the subscript indicates the number of pops that are performed, limited to $K = 10$ in our experiments. In Figure 3, we show the number of times each stack state has the highest probability on the Wik-2 test set (left figure, red denotes STAY and blue denotes PUSH) and the maximum action probability (right figure). The LSTM hidden states are not pushed onto the stack most of the times. We inspect what is pushed and find that the model uses the stack to mostly store hidden states for difficult predictions (more details below). While the overall distribution of the maximum action probability is not very peaky since the stack is not used

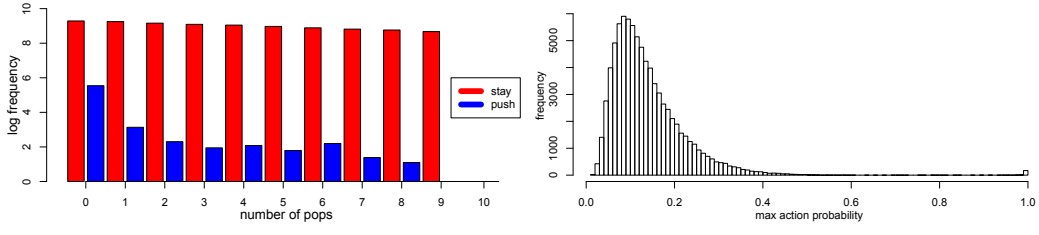

(a) Number of maximum action probability      (b) Value of maximum action probability

Figure 3: Statistics of the stack actions on Wik-2 test dataset. The left plot shows the number of times (in log space) each stack state has the highest probability. The $x$-axis represents the number of pops, and the color represents the last action taken after POP (red represents STAY, blue represents PUSH). We can see that most words are not pushed onto the stack and that the model takes advantage of the flexibility of the stack to pop multiple times. The right plot shows the value of the maximum action probability for all timesteps. We observe that for most words the maximum action probabilities are not peaky (the distribution is roughly uniform across all stack states). They tend to be peaky only when the model wants to use the stack.

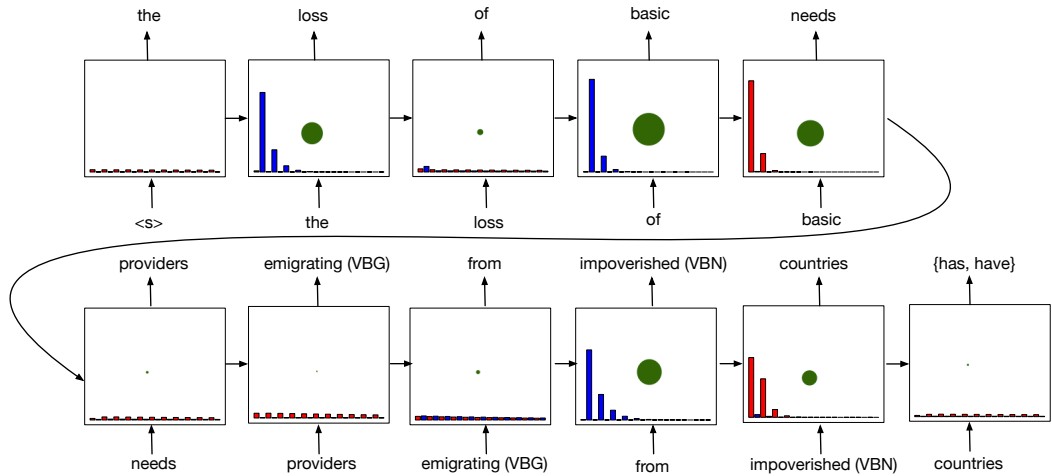

Figure 4: An illustration of how the stack memory is written and read for a correct prediction. We follow Linzen et al. (2016) and convert some words (e.g., emigrating, impoverished) to their part of speech tags–given inside the brackets in the example above—to limit the vocabulary size.

most of the times, we also observe that the action probability is relatively peaky when the stack is activated.

We next investigate how the stack improves accuracy on the Linzen dataset by looking into how it operates when making both a correct prediction and a wrong prediction. In Figure 4, we show a randomly selected test sentence *the loss of basic needs providers VBG from VBN countries {has, have}*. Similar to Figure 3, the red and blue bars represent STAY and PUSH, while the $x$-axis denotes the number of POPs. The $y$-axis, on the other hand, denotes the probability of each action. The green dots above each word shows the magnitude of the norm of the contribution of the memory vector $\mathbf{W}_{h,m}\mathbf{m}_t$ to the final hidden state $\tilde{\mathbf{h}}_t$ (bigger dots represent higher magnitudes). Interestingly, the model seems to use and push onto the stack when the next word prediction has a high entropy. For example, after the word *the*, *of*, or *from*, the model decides to increase its capacity by pushing the current hidden state onto the stack and activates its memory component (as illustrated by bigger green dots). In §2.1, we discuss connections of our stack model to adaptive computation time and variable computation RNN, which are designed to explicitly increase their capacity for difficult predictions. Our analysis shows that our stack model also exhibits this kind of behavior. For the Linzen dataset, we conjecture that the stack model is able to perform the best because it uses the stack as a controller to allow the sequential memory component from its base LSTM to carry longer term dependencies (e.g., tracking the subject of the sentence).

For comparison, we also show how the stack operates when it makes an incorrect prediction in Figure 5. Here, the stack behaves similarly, being mostly active for higher entropy predictions, although the model was unable to predict the correct verb right after *walls*.

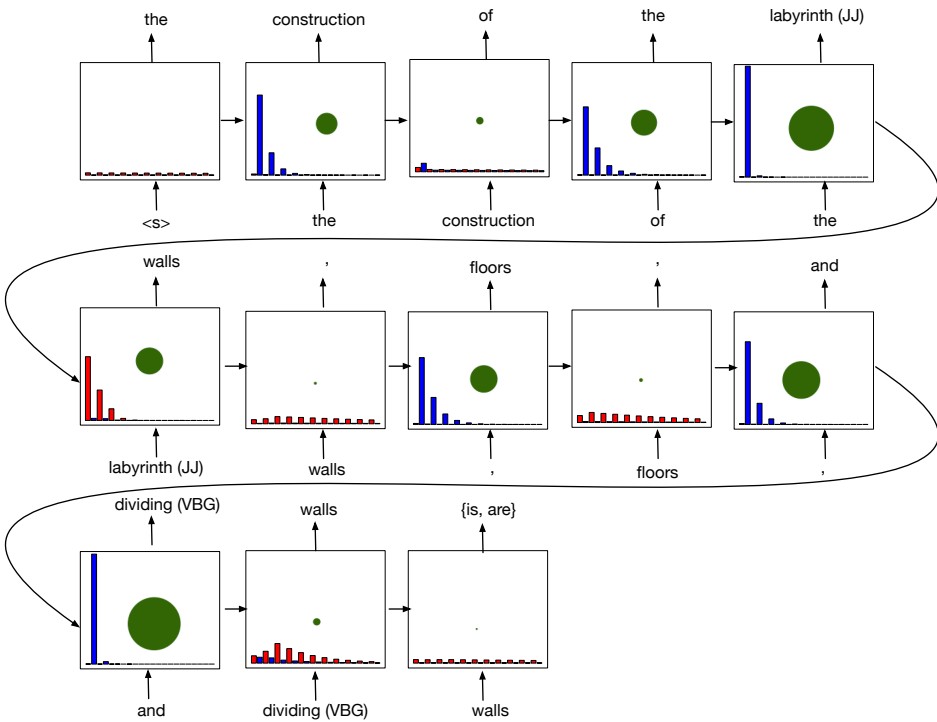

Figure 5: An illustration of how the stack memory is written and read for an incorrect prediction.

## 4 CONCLUSION

We proposed a generalization of the continuous stack model that allows a variable number of pop operations, and compared sequential, random access, and stack memory architectures for recurrent neural network language models. Our experiments on PTB and Wik-2 showed that adding the stack memory eliminates the need to tune window size in the random access attention model to achieve the best perplexity. We also evaluated these models on the Linzen syntactic dependencies dataset and demonstrated that the stack augmented model outperforms other methods in terms of both accuracy and perplexity, especially as the number of syntactic attractors increases.

ACKNOWLEDGEMENTS

The authors thank Edward Grefenstette for valuable feedback on an earlier draft of this paper, Tal Linzen for his assistance with the Linzen dataset, and the DeepMind language group for helpful discussions.

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
