# OpenReview forum: "Memory Architectures in Recurrent Neural Network Language Models"
_ICLR.cc/2018/Conference — Accept (Poster)_

### Official Review · AnonReviewer1 · 2017-11-27

**Rating:** 6
**Confidence:** 3

**Review:**

The authors propose to compare three different memory architecture for recurrent neural network language models:
vanilla LSTM, random access based on attention and continuous stack. The second main contribution of the paper is to propose an extension of continuous stacks, which allows to perform multiple pop operations at a single time step.
The way to do that is to use a similar mechanism as the adaptive computation time from Graves (2016): all the pop operations are performed, and the final state of the continuous stack is weighted average of all the intermediate states. The different memory models are evaluated on two standard language modeling tasks: PTB and WikiText-2, as well as on the verb number prediction dataset from Linzen et al (2016). On the language modeling tasks, the stack model performs slightly better than the attention models (0-2 ppl points) which performs slightly better than the plain LSTM (2-3 ppl). On the verb number prediction tasks, the stack model tends to outperforms the two other models (which get similar results) for hard examples (2 or more attractors).

Overall, I enjoy reading this paper: it is clearly written, and contains interesting analysis of different memory architecture for recurrent neural networks. As far as I know, it is the first thorough comparison of the different memory architecture for recurrent neural network applied to language modeling. The experiments on the Linzen et al. (2016) dataset is also interesting, as it shows that for hard examples, the different models do have different behavior (even when the difference are not noticeable on the whole test set).

One small negative aspect of the paper is that the substance might be a bit limited. The only technical contribution is to merge the ideas from the continuous stack with the adaptive computation time to obtain the "multi-pop" model. In the experimental section, which I believe is the main contribution of the paper, I would have liked to see more "in-depth" analysis of the different models. I found the experiments performed on the Linzen et al. (2016) dataset (Table 2) to be quite interesting, and would have liked more analysis like that. On the other hand, I found Figures 2 or 3 not very informative, as it is (would like to see more). For example, from Fig. 2, it would be interesting to get a better understanding of what errors are made by the different models (instead of just the distribution).

Finally, I have a few questions for the authors:
- In Figure 1. shouldn't there be an arrow from h_{t-1} to m_t instead of x_{t-1} to m_t?
- What are the equations to update the stack? I assume something similar to Joulin & Mikolov (2015)?
- Do you have any ideas why there is a sharp jump between 4 and 5 attractors (Table 2)?
- Why no "pop" operations in Figure 3 and 4?

pros/cons:
+ clear and easy to read
+ interesting analysis
- not very original

Overall, while not groundbreaking, this is a serious paper with interesting analysis. Hence, I am weakly recommending to accept this paper.

---

> ### Author Response · Authors · 2017-12-13
> **re: review**
>
> Thank you for your thoughtful review. Based on your suggestion, we have added examples of mistakes made by competing models in the Linzen experiment instead of just the Venn diagram (Table 3).
>
> Answers to your specific questions:
> - In Figure 1. shouldn't there be an arrow from h_{t-1} to m_t instead of x_{t-1} to m_t?
> Thanks for pointing this out. You are correct, we have updated the figure to fix the arrow.
>
> - What are the equations to update the stack? I assume something similar to Joulin & Mikolov (2015)?
> The equation to update the stack is given in Equation 1 (page 4).
>
> - Do you have any ideas why there is a sharp jump between 4 and 5 attractors (Table 2)?
> We think that there are two main reasons that could explain the sharp jump.
> The first one is because there are much fewer test examples in the dataset with 5 attractors (~150) compared to 4 attractors and above (400, 1100, 3800, ...), so the standard error on the reported accuracy is also higher (e.g., 91.6 +/- 1.2 and 88.0 +- 2.6 for the stack model with 4 and 5 attractors respectively).
> Another reason could be that sentences with more attractors are much longer than sentences with fewer attractors, so the difficulty increases non linearly as the number of attractors increases.
>
> - Why no "pop" operations in Figure 3 and 4?
> The "pop" operations are shown in the x axis (number of pops). Each pair of red and blue bars represents a single pop number.

---

### Official Review · AnonReviewer3 · 2017-11-28
**The authors propose a new stack augmented recurrent neural network, which supports continuous push, stay and a variable number of pop operations at each time step. They need to test the model on large corpora.**

**Rating:** 5
**Confidence:** 5

**Review:**

The authors propose a new stack augmented recurrent neural network, which supports continuous push, stay and a variable number of pop operations at each time step. They thoroughly compare several typical neural language models (LSTM, LSTM+attention mechanism, etc.), and demonstrate the power of the stack baed recurrent neural network language model in the similar parameter scale with other models, and especially show the superiority when the long-range dependencies are more complex in NLP area.

However the corpora they choose to test the ideas, are PTB and Wikitext-2, they're quite small, so the variance of the estimate is high, similar conclusions might not be valid on large corpora such as 1B token benchmark corpus.

Table 1 only gives results with the same level of parameters, the ppls are worse than some other models. Another angle might be the proposed model use the similar size of hidden layer 1500 plus the stack, and see how much ppl reductions it could get.

Finally the authors should do some experiments on machine translation or speech recognition and see whether the model could get performance improvement.

---

> ### Author Response · Authors · 2017-12-13
> **re: review**
>
> We would like to note that the main goal of the paper is to compare different memory architectures for RNN language models and analyze what kind of dependencies these models fail to learn.
>
> Perplexity is one metric to evaluate such models, so we use PTB and Wikitext-2---the two most commonly used language modeling datasets---to both compare these models and show that the memory models we implemented perform reasonably well compared to other work on these datasets.
>
> However, as noted in our paper, the overall perplexity on these datasets is strongly dominated by words that have few if any long term dependencies, making it difficult to assess when memory helps using perplexity alone.
> Instead of running these models on 1B corpus, which would have the same problem, we chose to include experiments on the Linzen dataset to be able to analyze these memory models further and get a better understanding of their strengths and limitations.
> We think this set of experiments adds more value and offers a more useful insight into memory augmented RNN LM than another opaque perplexity result on a larger corpus.
>
> Applications to machine translation and speech recognition are beyond the scope of this paper.

---

### Official Review · AnonReviewer2 · 2017-11-30
**Nice contribution to memory augmented recurrent neural network**

**Rating:** 8
**Confidence:** 5

**Review:**

The main contribution of this paper are:
(a) a proposed extension to continuous stack model to allow multiple pop operation,
(b) on a language model task, they demonstrate that their model gives better perplexity than comparable LSTM and attention model, and
(c) on a syntactic task (non-local subject-verb agreement), again, they demonstrate better performance than comparable LSTM and attention model.

Additionally, the paper provides a nice introduction to the topic and casts the current models into three categories -- the sequential memory access, the random memory access and the stack memory access models.

Their analysis in section (3.4) using the Venn diagram and illustrative figures in (3), (4) and (5) provide useful insight into the performance of the model.

---

### Public Comment · (anonymous) · 2017-11-21
**REINFORCE reward**

Could you please elaborate on the reward that was used for REINFORCE in the Single Computation Discrete Stack?

---

> ### Author Response · Authors · 2017-11-22
> **re: REINFORCE reward**
>
> The reward used is the log probability of the sequence generated, conditional on the sampled stack control decisions. This is thus optimizing an EM-like bound on the marginal likelihood.

---

### Public Comment · ~Tristan_Deleu1 · 2017-12-05
**ICLR 2018 Reproducibility Challenge**

As part of the ICRL 2018 Reproducibility Challenge, we are trying to replicate some of the experiments reported in this paper. We would like to contact the authors of this paper to discuss some of the technical details of the proposed model. We would be very grateful if the authors could get in touch with us at __________@_________.__

---

> ### Author Response · Authors · 2017-12-13
> **reproducibility**
>
> email sent! :)

---

### Decision · Program_Chairs · 2018-01-29
**ICLR 2018 Conference Acceptance Decision**

**Decision:**

Accept (Poster)

**Comment:**

This paper provides a comparison of different types of a memory augmented models and extends some of them to beyond their simple form. Reviewers found the paper to be clearly written, saying it "nice introduction to the topic" and noting that they "enjoyed reading this paper". In general though there was a feeling that the "substance of the work is limited". One reviewer complained that experiments were limited to small English datasets PTB and Wikitext-2 and asked why they didn't try "machine translation or speech recognition". (The author's note that they did try the Linzen dataset, and while the reviewers found the experiments impressive, the task itself felt artificial) . Another felt that the "multipop model" alone was not too large a contribution. The actual experiments in the work are well done, although given the fact that the models are known there was expectation of "more "in-depth" analysis of the different models". Overall this is a good empirical study, which shows the limited gains achieved by these models, a nevertheless useful piece of information for those working in this area.